# Depression and anxiety and their associated factors among caregivers of children and adolescents with epilepsy in three selected hospitals in Amhara region, Ethiopia: A cross-sectional study

**Mekonnen Tsehay** [1], **Mogesie Necho** [1], **Asmare Belete** [1], **Mengesha Srahbzu** [2]*

**1** Department of Psychiatry, College of Medicine and Health Science, Wollo University, Dessie, Amhara Regional State, Ethiopia, **2** Department of Psychiatry, School of Medicine, College of Medicine and Health Science, University of Gondar, Gondar, Amhara Regional State, Ethiopia

These authors contributed equally to this work.

* mengusew@gmail.com

## Abstract

### Background

The prevalence rates of depression and anxiety are unforeseen among primary caregivers of patients with epilepsy. Little attention is being given to the problem in Ethiopia.

### Objectives

This study aimed to assess the prevalence and associated factors of depression and anxiety among caregivers of children and adolescents with epilepsy in three selected hospitals in Amhara region, Ethiopia.

### Methods

Institution-based cross-sectional study was conducted in Ethiopia from January 1-30/2021. Systematic sampling technique was used. The Public Health Questionnaire (PHQ-9) and Generalized Anxiety Disorder (GAD-7) questionnaires were used to measure depression and anxiety respectively. Binary logistic regression model was employed independently for both depression and anxiety. Variables with P-values <0.2 were taken to multivariate analyses. Variables with P-value <0.05 in the multivariate analyses were considered to have a statistical association with depression and anxiety.

### Result

A total of 383 participants involved yielding a response rate of 90.5%. The prevalence of depression and anxiety were found to be 13.7% and 10.4% respectively. Being female (Adjusted Odds Ratio (AOR) = 1.21: 95% Confidence Interval (CI): 1.00, 3.82), being unmarried (AOR = 1.31; 95%CI: 0.32, 5.023), having history of chronic medical illness (AOR

**Data Availability Statement:** All relevant data are within the article and its Supporting Information files.

**Funding:** The author(s) received no specific funding for this work.

**Competing interests:** The authors have declared that no competing interests exist.

= 1.46; 95%CI: 1.07, 1.98), current seizure attack (AOR = 4.19; 95%CI: 1.36, 12.97), duration of care 6-11years (AOR = 1.80; 95%CI: 1.11, 7.58), duration of care > 11years (AOR = 6.90; 95%CI: 1.56, 30.49), moderate social support (AOR = 0.37; 95%CI: 0.13, 0.81), strong social support (AOR = 0.61; 95%CI: 0.22, 1.67) and currently use substance (AOR = 2.01;95%CI: 1.63, 6.46) were factors associated with depression. On the other hand, being unmarried (AOR = 1.47; 95%CI: 1.12, 1.93), current seizure attack (AOR = 1.81 with 95% CI = 1.28–2.54), able to read and write (AOR = 0.33; 95%CI: 0.14, 0.77), completed primary and secondary education (AOR = 0.54; 95%CI: 0.39, 0.76), current substance use (AOR = 1.466; 95%CI: 1.12, 1.93), being parent (AOR = 2.55; 95%CI: 1.31, 4.96), rural (AOR = 3.75; 95%CI: 1.40, 10.04) and grand mal type (AOR = 2.21; 95%CI: 1.68, 2.91) were factors associated with anxiety.

## Conclusions

In our study, approximately one in fifteen and more than one in ten caregivers had depression and anxiety respectively. The result of this study suggested that healthcare providers need to pay more attention to the psychological well-being of all caregivers of children and adolescents with epilepsy.

## Introduction

Numerous neurological problems disturb the person's working condition and result in enduring incapacity. According to the World Health Organization (WHO) report, epilepsy and other neurological problems remain among the top public health concerns. It has been revealed that about one billion people are suffered from these health problems globally [1]. The impact of neurological health problems is estimated to be 6% of the global burden of disease, and their significance will endure growing with age globally [2].

The symptoms and disabilities associated with neurological conditions have a major impact on individuals, their families and caregivers, and health service use [1, 3]. Individuals living in the community with neurological conditions receive the majority of their care from primary caregivers, such as family and friends. However, when neurological conditions worsen over time, they produce a variety of symptoms and functional impairments that often increase demands on primary caregivers [4].

Childhood epilepsy affects the family significantly. The condition is associated with a range of cognitive and behavioral difficulties that are often unrecognized and under-treated, and have a significant effect on health-related quality of life [5]. Difficulties in school include increased risk of academic underachievement and school attendance problems, which may also affect parental wellbeing [6].

Caregiver psychological distress increases as the amount of time spent providing care to recipients with neurological conditions increases [7]. In general, individuals prefer to remain living at home in the community, but to accommodate that preference attention to the primary caregiver is crucial, as caregiver distress may lead to depression, anxiety, reduced quality of care, and the recipient's ability to remain in the community [8–10]. The psychological distress experienced by caregivers can affect the quality of care provided for these children and adolescents, which ultimately can affect the prognosis of these patients [9, 10].

A review study conducted in Ethiopia has shown the prevalence of anxiety among caregivers of children and adolescent with epilepsy in different countries ranges from 55–58% [9]. A study conducted in America reported prevalence of anxiety among parents of children and adolescents with epilepsy to be 45% [7]. Another study conducted in Sweden among parents of children with drug-resistant epilepsy who were referred for pre-surgical evaluation reported as parents had 'possible/probable' anxiety (52% vs. 38%) even though it is almost similar for depression (30% vs. 22) [11]. A study conducted in Canada among parents of youth aged 6–17 years with epilepsy also reported the prevalence of moderate/severely depression to be 25% [12].

Data from Africa, Nigeria shows that the prevalence of anxiety among the caregivers was 12%, while that of depression was 50.5% in Nigeria [13].

Despite being aware about the situation of depression and anxiety among caregivers of patients with epilepsy in different settings, unfortunately we know little about the prevalence and associated factors of depression and anxiety among caregivers of children and adolescents with epilepsy in relation to their caregiving role in Ethiopia. Therefore, this study aimed to determine prevalence and associated factors of depression and anxiety among caregivers of children and adolescents with epilepsy at Dessie referral hospital, Bahr Dar referral Hospital, and Gondar university hospital, Amhara region, Ethiopia.

## Methods and materials

### Ethics statement

Ethical clearance was obtained from Wollo University health science college Institutional Research Ethics Review Committee. Formal letter of permission was obtained from the three hospital administrations. Written Informed consent was obtained from each participant. The aim of the study and possible risks and benefits were explained to participants. Participants were also been informed that they have the right not to participate in the study or to stop at any time in between or not to answer any questions they were not willing to answer. Confidentiality was maintained at all times and no unauthorized person had access to the information obtained from the clients.

### Study design, period and setting

Institutional based cross-sectional study was conducted from January 1–30, 2021. The study was conducted at three hospitals of the Amhara region (Dessie referral hospitals, Felege-Hiwot referral hospitals, and Gondar university hospital). All three hospitals have neurologist, and the neurology outpatient departments provide clinical service for more than 1227 (112, 225 and 890 respectively) people in their respective catchment areas. They are also providing organized psychiatric inpatient and outpatient services which most of the time serve for epileptic patients specially when comorbid with mental illness.

### Study population

All Caregivers of child and adolescents with epilepsy who had follow-up at Neuropsychiatric Case Team of Dessie referral hospital, Felege-Hiwot referal Hospitals, and Gondar university hospital during the study periods were considered as a study population. Caregivers who were 18 years old and above who had been providing care for more than 6 months for child and adolescents diagnosed with epilepsy were included in the study. Caregivers who have no direct involvement in providing care, those with history of known psychiatric disorder before being a caregiver, and those who are unable to hear or speak were excluded from the study.

## Sample size determination and sampling procedure

The sample was determined by using single population proportion formula considering the following assumptions 95% of confidence interval. There was no local or national data on the prevalence of General anxiety disorders among caregivers of child and adolescents with epilepsy. Hence, prevalence of 50% and marginal error (d) of 5% were used to maximize sample size. Considering 10% for nonresponse rate, the final sample size was taken to be 384+39 = 423.

We used the systematic random sampling technique to select 423 primary caregivers of children and adolescents with epilepsy having follow-up for the treatment of their children and adolescents with epilepsy from any of the three selected hospitals after proportional allocations to population size of hospitals.

## Data collection instruments

The outcome variables were depression and anxiety. Depression was measured by using an interviewer-administered PHQ-9. PHQ-9 score ranges from 0 to 27. Each of the 9 items was scored from 0 ("not at all") to 3 ("nearly every day"). A PHQ-9 score 10–14, 15–19 and 20–27 indicates moderate, moderately severe, and severe depression which requires immediate initiation of therapy. Moreover, PHQ-9 has been validated in Ethiopian healthcare context with specificity and sensitivity of 67 and 86% respectively. A cut-off point of 10 and above has been used to screen depression [14]. In the current study, Cronbach's alpha of the scale was 0.79.

Anxiety was measured using GAD-7, screening and diagnostic tool. This assessment scale was adapted to evaluate the presence of anxiety among caregivers. The 7 items were scored from 0 (not at all) to 3 (nearly every day) with an overall GAD-7 scale score ranging from 0–21. The scale presents a rapid, efficient, reliable, and valid method for detecting the presence of a common anxiety disorder. A score 0–4 represents minimal anxiety, a score of 5–9: mild anxiety, a score of 10–14: Moderate Anxiety and a score greater than 15: severe Anxiety. A cut-off point at a score of 10 and above on the GAD-7 scale had been defined as anxiety with a sensitivity and specificity of 89% and 82% respectively [15].

Perceived social support was among the psychological factors considered in this study, and it will be assessed by using the Oslo 3-item social support scale which had a sum score ranges from 3 to 14 and had three broad categories. According to this category, respondents who scored 3–8, 9–11 and 12–14 was considered as having poor, moderate and strong social support respectively [16]. Chronic medical illness and family history of mental illness were assessed by using yes/no questions in primary caregivers of children and adolescents with epilepsy.

## Operational definition

**Depression.**  According to PHQ-9, scores 10–14, 15–19 and 20–27 indicates moderate, moderately severe, and severe depression respectively [14].

**Anxiety.**  According to GAD-7 screening and diagnostic tool, a participant who score <5, 6–9 and 10–15 indicates mild, moderately, and severe Possible diagnosis of GAD respectively [15].

**Social support.**  According to the Oslo 3-item social support scale, respondents who scored 3–8, 9–11 and 12–14 was considered as having poor, moderate and strong social support respectively [16].

## Data collection procedures

Initially, all questionnaires were translated into local language (Amharic) before data collection and translated back by another bilingual expert in both English and Amharic to check its

consistency. Data was collected from primary caregivers accompanied with children and adolescents with epilepsy who had follow-up treatment service using interview technique at child and adolescent neurologic clinic in Dessie, Felegehiwot referral hospital and Gondar university hospitals. Pre-test was done on a sample (5% of the total sample) of primary caregivers of children and adolescents with epilepsy attending outpatient clinic at Dessie referral hospital prior to data collection will be implemented. The finding of the pretest was not included in the main research report. Clinicians working in child and adolescent neurologic clinics will link caregivers with data collectors, and the data collectors interviewed primary caregivers who are eligible. Training was given to four data collectors and two supervisors on basic data collection and interview techniques for each hospital. Data quality and its completeness were monitored by supervisors at daily basis.

## Data processing and analysis

Data was coded and entered into the Epi-data software version 3.1, and exported to Statistical Package for Social Science (SPSS, version 21) for analysis. After data cleaning, bivariate analysis were used to assess the associations between dependent and independent variables. Adjusted odds ratio with a 95% confidence interval will be used to estimate the strength of the association. All variables with p-value of less than 0.2 in the bivariate logistic regression were further analyzed using multivariate logistic regression analyses to control the confounding effects. Variables with a P-value less than 0.05 in the multivariate logistic regression were declared to be significantly associated with depression and anxiety.

## Result

### Socio-demographic characteristics of caregivers of children and adolescents with epilepsy

A total of 383 participants were enrolled in the study which resulted in an overall response rate of 90.5%. 282 (73.6%) were male. The mean age of the respondents was 39.59 (Standard deviation (SD) = ±9.97) years. Most of the participants 302 (78.9%), were married and about 251 (65.5%) were from rural areas (**Table 1**).

### Socio-demographic characteristics of child and adolescents with epilepsy

The mean age of the children was 10.4 with an SD of 4.04, and 53% were males. 31.9% were ill for more than five years. Majorities (73.9%) of children and adolescents have a Tonic-clonic type of seizure. and 82.5% have controlled seizures (**Table 2**).

### Prevalence of depression and anxiety

The Prevalence of depression among caregivers was 13.70% with a 95% CI (5.72, 18.40). 7.82%, 4.42%, and 1.51% had reported moderate, moderately severe and severe depression symptoms according to the PHQ-9 severity score. The mean score of participants on PHQ-9 was 8.85 with an SD of ± 4.91. On the other hand, the prevalence of anxiety among participants was 10.4% with 955 CI (5.74, 14.81). Of these 6% have moderate and only 4.4% had severe anxiety according to the GAD-7 rating scale. The mean score of participants on GAD-7 was 6.11 with an SD of ±3.72.

### Factors associated with depression among participants

Both binary and multiple logistic regression analysis models were done separately for depression and anxiety on socio-demographic, clinical, and behavioral variables.

**Table 1. Socio-demographic characteristics of caregivers of children and adolescents with epilepsy in selected three hospitals of Amhara region, Ethiopia.**

| Variables | Sub-type | Frequency/mean | Percent (%)/SD |
|---|---|---|---|
| Age(years) | 18–30 | 87 | 22.7 |
| | 31–40 | 144 | 37.6 |
| | 41–50 | 90 | 23.5 |
| | >50 | 62 | 16.2 |
| | Mean | 39.59 | 9.97 |
| sex | Male | 282 | 73.6 |
| | Female | 101 | 26.4 |
| Marital status | Married | 302 | 78.9 |
| | Single | 58 | 15.1 |
| | Divorced | 4 | 1.0 |
| | widowed | 12 | 3.1 |
| | separated | 7 | 1.8 |
| Religion | Orthodox | 352 | 91.9 |
| | Muslim | 12 | 3.1 |
| | Protestant | 19 | 5.0 |
| Educational status | Illiterate | 164 | 42.8 |
| | Read and write | 117 | 30.5 |
| | Primary or secondary | 67 | 17.5 |
| | College and above | 30 | 7.8 |
| Occupational status | Government employed | 40 | 10.4 |
| | NGO employed | 26 | 6.8 |
| | merchant | 127 | 33.2 |
| | Farmer | 142 | 37.1 |
| | House wife | 17 | 4.4 |
| | Student | 18 | 4.7 |
| | Others [A] | 13 | 3.4 |
| Number of family | six and above | 11 | 2.9 |
| | three to five | 221 | 57.7 |
| | two or one | 151 | 39.4 |
| Number of children with epilepsy | One | 299 | 78.1 |
| | Two | 84 | 21.9 |
| History of Chronic medical illness | Yes | 12 | 3.1 |
| | No | 371 | 96.9 |
| Caregiver epilepsy status | Yes | 23 | 6.0 |
| | No | 360 | 94.0 |
| Duration of care | 5 years and less | 274 | 71.5 |
| | 6–10 years | 89 | 23.2 |
| | 11 and above years | 20 | 5.2 |
| | Mean | 4.45 | 3.03 |
| Relationship with child or adolescents | Father | 229 | 59.8 |
| | Mother | 102 | 26.6 |
| | sister or brother | 52 | 13.6 |
| Residence | Urban | 132 | 34.5 |
| | Rural | 251 | 65.5 |

Note: Others [A] = self-employed, Labor worker, Jobless.

**Table 2. Socio-demographic characteristics of children and adolescents with epilepsy in selected three hospitals of Amhara region, Ethiopia.**

| Variable | Sub-variable | Frequency/mean | %/ SD |
|---|---|---|---|
| Age | 5 and less | 72 | 18.8 |
| | 6–10 | 100 | 26.1 |
| | 11–15 | 174 | 45.4 |
| | 14 and above | 37 | 9.7 |
| | Mean | 10.4 | 4.04 |
| sex | Male | 203 | 53.0 |
| | Female | 180 | 47.0 |
| Educational status | Illiterate | 185 | 48.3 |
| | read and write | 167 | 43.6 |
| | primary or secondary school | 31 | 8.1 |
| History Comorbid medical illness | Yes | 27 | 7.0 |
| | No | 356 | 93.0 |
| History of Head injury due epilepsy | Yes | 18 | 4.7 |
| | No | 365 | 95.3 |
| Type of anti-epileptic drug | carbamazepine | 28 | 7.3 |
| | valproate | 37 | 9.7 |
| | phenytoin | 233 | 60.8 |
| | phenobarbital | 85 | 22.2 |
| Duration of illness | 5 years and less | 261 | 68.1 |
| | 6–10 years | 93 | 24.3 |
| | 11 and above years | 29 | 7.6 |
| | Mean | 4.45 | 3.03 |
| Type of seizure | Convulsive | 304 | 79.4 |
| | Non-convulsive | 79 | 20.6 |
| Frequency of seizure attack before AED within a month | 2 times | 89 | 23.2 |
| | 3–4 times | 143 | 37.3 |
| | Above 4 times | 151 | 39.4 |
| Frequency of seizure attack in the last month | None | 316 | 82.5 |
| | 1–2 times | 43 | 11.2 |
| | 3 and 4 times | 7 | 1.8 |
| | Above 4 times | 17 | 4.4 |

For both models (depression and anxiety) variables with a p-value of less than 0.2 in binary logistic regression were taken into multivariable logistic regression. In the depression multivariable logistic regression model sex, marital status, a history of chronic medical illness, duration of care, current seizure attack, current substance use, and level of social support availability were found associated with depression symptoms among caregivers of children and adolescents. The odds of developing depression among female participants were found to increase by 1.21 when compared to male participants. The odds of developing depression in unmarried caregivers increased by 1.31 when compared to married caregivers. Having history of chronic medical illness, current seizure attack and current substance use increase the odds of developing depression by 1.46, 4.19, and 2.01 respectively when compared to their counterparts. Duration of care is also another significant risk factor for developing depression in which those with a duration of care was between 6 and 11 and greater than 11 increased the odds of developing depression by 1.80 and 6.90 respectively when compared to those with a duration of care less than or equals to 5 years. Social support was found to be another significantly associated factor with depression in which the odds of developing depression among

**Table 3. Bivariate and multivariate analysis of factors associated with depression among caregivers of children and adolescents with epilepsy attending follow up visits in the three hospitals (n = 383).**

| Variables | Sub category | Depression | |
| --- | --- | --- | --- |
| | | COR (95% CI) | AOR (95% CI) |
| Sex | Male | 1 | 1 |
| | Female | 1.81(1.11, 3.54) | **1.21(1.01, 3.82)** |
| Marital status | Married | 1 | 1 |
| | unmarried | 2.07 (1.03, 6.90) | 1.31(0.32, 5.02) |
| History of Chronic medical illness | No | 1 | 1 |
| | Yes | 2.48(1.44, 4.28) | 1.46(1.07, 1.98) |
| Duration of care | 5 years and less | 1 | 1 |
| | 6–10 years | 5.63(1.93, 16.44) | **1.80(1.11, 7.58)** |
| | 11 and above years | 16.62(5.61, 49.26) | **6.90(1.56, 30.49)** |
| Current of seizure attack in the last month | No | 1 | 1 |
| | Yes | 1.39(1.01, 2.04) | 4.19(1.36, 12.97) |
| At least one Current substance use | No | 1 | 1 |
| | Yes | 2.02 (1.29, 3.06) | **2.01 (1.63–6.46)** |
| Level of social support based on Oslo-3 | poor social support | 1 | 1 |
| | moderate social support | 0.23(0.07, 0.74) | **0.37(0.13, 0.81)** |
| | strong social support | 0.67(0.30, 1.91) | 0.61(0.22, 1.67) |

Note: unmarried = single, divorced widowed, and separated; COR = Crude Odds Ratio; AOR = Adjusted Odds Ratio. Model fitness = Hosmer-lemeshow (p-value = 0.73).

those with moderate social support decreased by 36.6% as compared to those with poor/low social support (**Table 3**).

## Factors associated with anxiety among participants

In the multivariable logistic regression model for anxiety, marital status, current seizure attack, educational status of caregivers, current substance use, relationship with child and adolescents, residence, and type of epileptic seizure of child and adolescents were found significantly associated with anxiety.

In this model, after we classified marital status into married and unmarried (single, divorced, widowed, and separated) the odds of developing anxiety among unmarried caregivers were found to be increased by 1.47 times. Having a current seizure attack, current substance use, being a parent by a relationship with children and adolescents with epilepsy and being from a rural residence increase the odds of developing anxiety symptoms by 1.81, 1.47, 2.55, and 3.75 respectively. The odds of developing anxiety are 2 times more likely among caregivers of children and adolescents with current convulsive seizures when compared to their counterparts. Unlike the depression model, in the anxiety model educational status of caregivers was found significantly associated. The odds of developing anxiety among those caregivers whose educational status was able to read and write and have completed primary and secondary school decreased by 32.8%, and 54.1% respectively when compared to those caregivers who can't read and write (**Table 4**).

## Discussion

This study has found a significant number of depression and anxiety among caregivers of children and adolescents with epilepsy. Different variabales like gender, duration of giving care,

**Table 4. Bivariate and multivariate analysis of factors associated with anxiety among caregivers of children and adolescents with epilepsy attending follow up visits in the three hospital (n = 383).**

| Variables | Sub category | COR(95% CI) | AOR(95% CI) |
|---|---|---|---|
| | **Anxiety** | | |
| Marital status | Married | 1 | 1 |
| | unmarried | 3.01(1.53, 5.90) | **1.47(1.12, 1.93)** |
| Current seizure attack in the last month | No | 1 | 1 |
| | Yes | 3.47 (2.07, 5.82) | **1.81(1.28, 2.54)** |
| Educational status | Illiterate | 1 | 1 |
| | Read and write | 0.38 (0.25, 0.58) | **0.33(0.14, 0.77)** |
| | Primary or secondary school | 0.73(0.58, 0.93) | **0.54 (0.39, 0.76)** |
| At least one current substance use | No | 1 | 1 |
| | Yes | 1.56(1.07, 2.27) | **1.47(1.12, 1.93)** |
| Relationship with child and adolescents | Other relatives | 1 | 1 |
| | Parent | 3.32 (1.78, 6.19) | 2.55 (1.31, 4.96) |
| Residence | Urban | 1 | 1 |
| | Rural | 5.63(1.93, 16.44) | **3.75 (1.40, 10.04)** |
| Type of epilepsy of child and adolescent | Convulsive | 3.62(1.62, 6.26) | **2.21 (1.68, 2.91)** |
| | Non-convulsive | 1 | 1 |

Note: unmarried = single, divorced widowed, and separated; COR = Crude Odds Ratio; AOR = Adjusted Odds Ratio. Model fitness = Hosmer-lemeshow (p-value = 0.78).

current substance use, and level of social support were variables found to have a significant association with depression. On the other hand variables like marital status, educational status, current seizure attack, current substance use residence, and type of seizure were factors found to have a significant association with anxiety.

This study revealed the prevalence of depression among caregivers of children and adolescents with epilepsy to be 13.7% with a 95% CI (5.72, 18.40). It is in line with a previous study conducted in Nepal 8.5% [17]. However, it is lower than study results reported in the USA, 45% [7], Canada, 25% [12], Nigeria, 50.5% [13] and Sweden which was reported to be 30% and 22% for possible and probable depression respectively [11]. The possible reason for the difference might be variation in the study population in which a study on Canada used specifically children with uncontrolled seizures [12], and patients presenting for the surgical procedures were assessed in Sweden [11]. Another possible reason might be the variation in sample size used between our study and previous studies.

This study also revealed that 10.4% of caregivers of children and adolescents with epilepsy presented symptoms of anxiety with 95% CI (5.74, 14.81). The result of this study on the prevalence of anxiety is in line with the results of previous studies in Nigeria, 12% [13], and Nepal, 7.5% [17]. However, the result of this study was lower than studies conducted in Sweden which reported possible/probable anxiety to be 52/38% among caregivers of children with epilepsy [11]. The reason behind this difference might be a difference in the study population in which children with epilepsy referred for presurgical evaluation were assessed in Sweden [11].

Among factors associated with depression, caregivers of children and adolescents with epilepsy are 1.47 times more likely to develop depression when compared to their counterparts. The increased risk might be rooted in the presence of low self-esteem to cope with caregiving-related stress when being unmarried and lack of social support specifically spousal support [18, 19]. Those caregivers of children and adolescents with current seizure attacks have 1.81 times increased risk of developing depression than their counterparts. This might be due to

the reason that patients with current seizure attacks must get close follow-up and care. Having spent a prolonged time with patients might lead them to distress [20]. Another possible explanation for this difference might be increased stress as a result of increased fear of being stigmatized [21].

The odds of developing depression among parental caregivers of children and adolescents with epilepsy are 2.55 times greater than those caregivers who are not their parents. These can be explained by the fact that there is an intimate emotional attachment between parent-son/daughter. This, in turn, leads parents to easily get into emotional turmoil when they look at the suffering of their son/daughter on a day-to-day basis [22]. Caregivers of children and adolescents with epilepsy who currently use substances are 1.47 times more likely to develop anxiety when compared to non-users. This might be because psychoactive substances alter the function of the central nervous system. This may alter the distribution of brain neurotransmitters which may also result in emotional disturbances [23, 24].

We have also found that caregivers who are from rural areas are 3.75 times more likely to develop depression when compared to those from urban areas. This might be due to socio-demographic differences like educational status, access to media for information regarding developing coping skills, and economic status [25]. The odds of developing depression among caregivers of children and adolescents with grand mal epilepsy are 2.21 times higher than those with other types of seizure disorder. This might be due to the catastrophic reactions of caregivers towards features of grand mal seizure attack and its periodical and frequent occurrence [26]. This might be due to common sleep cycle disturbance in patients with grand mal epilepsy and the increased burden of caring for them [27]. The odds of developing depression among caregivers of children and adolescents with epilepsy decreased by 67.2% and 61.3% among those who can read and write and completed primary and secondary school respectively when compared to those who can't read and write. This might be because stress might get decreased when there is better awareness and acquire strong coping skills through educational exposure [28].

Regarding factors associated with anxiety, being a female caregiver is 1.21 times more likely to develop anxiety when compared to a male. This is supported by previous study that reported higher anxiety symptoms among female caregivers [29]. This might be because female caregivers have more caregiving burden which in turn leads them to have increased risks for poor psychosocial and physical wellness [30]. It might also be due to females might not have good coping skills for the increased burden of caregiving as men [31]. The risk of developing anxiety among unmarried caregivers of children and adolescents with epilepsy increases by 30.5%. The possible reason behind this might be the fact that there is increased stress among caregivers when taking the role alone due to a higher perceived caregiving burden [29, 32, 33]. This might also be due to the absence of spousal support in fighting against being stigmatized which may lead them to emotional disturbance [32, 33].

Having a history of chronic medical illness is 1.46 times more likely to develop anxiety among caregivers of children and adolescents with epilepsy when compared to their counterparts. This might be due to the increased risk of having a mental illness when biological etiologies come with caregiving related stress. Another possible reason might be the fact that patients who are giving care to their relatives are less likely to follow their medical treatment which leads them to poor physical health including mental health [34, 35].

Caregivers of children and adolescents with current seizure attacks are 4.192 times more likely to develop anxiety when compared to their counterparts. This might be due to the possible stress of being with the patient for a prolonged time for close follow-up [20]. This study also revealed that caregivers of children and adolescents with epilepsy who are currently using sunstances are 2.012 times more likely to develop anxiety than the non-users. The increased

risk of anxiety might be due to the fact that psychoactive substances alter the function of the central nervous system. This is may have disturbance in brain neurotransmitters which may also result in emotional disturbances [23, 24].

Furthermore, this study revealed as the longer the period of caregiving the higher the risk of developing anxiety among study participants. This might be due to the fact that a longer period of caregiving might leads to greater financial expenditure for medical services and poor health-related quality of life which might contribute to developing anxiety [36, 37]. The risk of developing anxiety among study participants who have moderate and social support is found to be decreased by 33.4% and 39.2% respectively. The possible reason for the difference might be the fact that people with social support can better cope with stress and reduce the risk of developing emotional disturbances [38, 39].

## Limitation of the study

This study might have some limitations. Because of the nature of the cross-sectional study, we can't establish a cause and effect relationship between depression and anxiety and associated factors. The study didn't measure the degree of caregiving burden among caregivers because of different parent-child relationships in the study participants.

## Conclusion and recommendation

In this cross-sectional study among caregivers of children and adolescents with epilepsy, more than one in ten and approximately one in fifteen caregivers had anxiety and depression respectively. The result of this study suggested that healthcare providers need to pay more attention to the psychological well-being of all caregivers of children and adolescents with epilepsy. Special attention should be given to caregivers of children and adolescents with current seizure attacks.

## Supporting information

**S1 Dataset.**
(SAV)

## Acknowledgments

We would like to thank three hospital administrators and students who gave us support and information. We also would like to thank data collectors for their genuine help, especially during the data collection and data entry of this study.

## Author Contributions

**Conceptualization:** Mekonnen Tsehay, Mogesie Necho, Asmare Belete, Mengesha Srahbzu.

**Data curation:** Mekonnen Tsehay, Mogesie Necho, Asmare Belete, Mengesha Srahbzu.

**Formal analysis:** Mekonnen Tsehay, Mogesie Necho, Asmare Belete, Mengesha Srahbzu.

**Investigation:** Mekonnen Tsehay, Mogesie Necho, Asmare Belete, Mengesha Srahbzu.

**Methodology:** Mekonnen Tsehay, Mogesie Necho, Asmare Belete, Mengesha Srahbzu.

**Project administration:** Mekonnen Tsehay, Mogesie Necho, Asmare Belete, Mengesha Srahbzu.

**Supervision:** Mekonnen Tsehay, Mogesie Necho, Asmare Belete, Mengesha Srahbzu.

**Validation:** Mekonnen Tsehay, Mogesie Necho, Asmare Belete, Mengesha Srahbzu.

**Visualization:** Mekonnen Tsehay, Mogesie Necho, Asmare Belete, Mengesha Srahbzu.

**Writing – original draft:** Mekonnen Tsehay, Mogesie Necho, Asmare Belete, Mengesha Srahbzu.

**Writing – review & editing:** Mekonnen Tsehay, Mogesie Necho, Asmare Belete, Mengesha Srahbzu.

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
