## [Decision Letter · Decision Letter 0]

17 May 2022

PONE-D-22-05990Depression and anxiety and their associated factors among care givers of children and adolescents with epilepsy in selected three hospitals of Amhara region, Ethiopia.PLOS ONE

Dear Dr. BIRESAW,

Thank you for submitting your manuscript to PLOS ONE. After careful consideration, we feel that it has merit but does not fully meet PLOS ONE’s publication criteria as it currently stands. Therefore, we invite you to submit a revised version of the manuscript that addresses the points raised during the review process. Please submit your revised manuscript by Jul 01 2022 11:59PM. If you will need more time than this to complete your revisions, please reply to this message or contact the journal office at plosone@plos.org. Please include the following items when submitting your revised manuscript:A rebuttal letter that responds to each point raised by the academic editor and reviewer(s). You should upload this letter as a separate file labeled 'Response to Reviewers'.A marked-up copy of your manuscript that highlights changes made to the original version. You should upload this as a separate file labeled 'Revised Manuscript with Track Changes'.An unmarked version of your revised paper without tracked changes. You should upload this as a separate file labeled 'Manuscript'.

We look forward to receiving your revised manuscript.

Kind regards,

Francesco Deleo, MD

Academic Editor

PLOS ONE

Journal Requirements:

2. Thank you for submitting the above manuscript to PLOS ONE. During our internal evaluation of the manuscript, we found significant text overlap between your submission and the following previously published works, some of which you are an author.

- https://link.springer.com/article/10.1186/s12913-015-1010-1?code=0c8847bb-9822-4492-a10d-a2f258e0ebe9&error=cookies_not_supported

Please revise the manuscript to rephrase the duplicated text, cite your sources, and provide details as to how the current manuscript advances on previous work. Please note that further consideration is dependent on the submission of a manuscript that addresses these concerns about the overlap in text with published work.

3. Please amend your current ethics statement to address the following concerns:

a) Did participants provide their written or verbal informed consent to participate in this study?

Reviewers' comments:

Reviewer's Responses to Questions

**Comments to the Author**

1. Is the manuscript technically sound, and do the data support the conclusions?

Reviewer #1: Yes

Reviewer #2: Partly

2. Has the statistical analysis been performed appropriately and rigorously? 

Reviewer #1: Yes

Reviewer #2: No

3. Have the authors made all data underlying the findings in their manuscript fully available?

Reviewer #1: Yes

Reviewer #2: Yes

4. Is the manuscript presented in an intelligible fashion and written in standard English?

Reviewer #1: No

Reviewer #2: Yes

5. Review Comments to the Author

Reviewer #1: The fact that this is an institution-based study among caregivers of patient with epilepsy at public hospitals is a great strength of the investigators' approach. While this is not clearly stated, and I strongly encourage the investigators to state this clearly and underscore the contribution of this study to the overall literature. focus on important elements of the methods section, grammar, and language use. Comments are provided from here with 1-16, look there all

1. Abstract background line 26, say depression and anxiety disorders are also today’s silent health crises in primary caregivers of children and adolescents with epilepsy? Tell as some thing that you consider silent health crisis unless you must rewrite and revise the statement as well.

2. Abstract method part line33-34, Binary logistic regression was employed to carry out bivariate and multivariate analyses were fitted independently for both depression and anxiety. please either separate or rewrite it again.

3. Why nonresponse rate is high? which is around 10%

4. Be consistent when you write a statement especially “depression and “anxiety” or anxiety and depression”. When we see the topic, it says Depression and anxiety and their associated factors but, in abstract part of result line 38, prevalence of anxiety and depression were found to be 10.4% and 13.7% respectively. Which one is correct?

5. Please define it what mean by unmarried? line 39

6. Similar with the above one “having history of chronic medical illness” who are they chronic medical illness? do they have followed up or simply report of care giver?

7. Do you have any suggestion why duration classification with this “duration of care 6-11years”?

8. Avoid such a kind of words” almost” rather put the number or percent alone.

9. Be consistent either use “general anxiety disorders” or “anxiety” line 96

10. Occupational status classification “other “is greater than housewife which is 8%, it is not advisable this number of individuals to put with term other rather put with their own classification or put remark below the table.

11. Line 211 what is the term” Gad” mean?

12. Line 213 “P-value “repeated two times.

13. In logistic regression the value after a point should be two digits.

14. On table -3 of logistic regression model, you must put the value of each variable respondents.

15. Please put confidence interval of the prevalence unless it is difficult to know and compare with others.

16. It is difficult to understand the English language

Reviewer #2: Dear authors, I revised with interest your manuscript entitled “Depression and anxiety and their associated factors among care givers of children and adolescents with epilepsy in selected three hospitals of Amhara region, Ethiopia” for PLOS ONE, where you present a cross-sectional analysis on the risk of developing anxiety and depression in relation to caregiving activity of children with epilepsy.

In general, the manuscript is well written: the analytic plan is simple but consistent with the aims and results are clearly presented. I have few comments for your which I hope you can exploit to enhance the overall quality of your paper.

1. In general, a revision on the use of English is needed in different sections of the manuscript: in some places, I had difficulties in understanding what you meant.

2. Please ensure that abbreviations are given in full at the first use, abstract included.

3. Lines 83-90: a revision on the use of English is essential

4. Lines 205-209: please give means and SD for both GAD-7 and PHQ-9.

5. Liens 211-258: these sections largely duplicate tables 3 and 4, and by this I mean that at least 50% of the text could be deleted. You have to reduce redundancy as it makes reading difficult: just focus on the results of the multivariable regression analysis and do not repeat OR values.

6. With regard to OR, please just refer to them as “OR” and not to AOR (I presume “adjusted” OR) or COR (unclear what does the C stand for).

7. Table 3 and 4: I do not see the utility of the “COR” column, and my suggestion is to delete it.

8. An important element that is missing here is a measure of model goodness. You have the OR, however how much of anxiety/depression is explained by your predictor. Instead of using Nagelkerke pseudo R-squared, please use the C-Statistic (i.e. the area under the receiver operating curve - ROC - for the predicted versus the actual data) used to assess the whole explanatory power of the model.

9. Also, please include the intercept in the model and provide the -2 log likelihood difference with chi-square to test the difference between the full model and the model based on the intercept only.

10. Please begin discussion with a paragraph stating the most relevant results. Do not repeat results (with data), but state what was found in “practical terms”. Once this is done, you can start discussing against relevant literature.

6. PLOS authors have the option to publish the peer review history of their article (what does this mean?). If published, this will include your full peer review and any attached files.

Reviewer #1: No

Reviewer #2: No

---

## [Author Response · Author response to Decision Letter 0]

7 Jun 2022

Response to reviewers

Title: “Depression and anxiety and their associated factors among care givers of children and adolescents with epilepsy in selected three hospitals of Amhara region, Ethiopia." 

Ref: Submission ID: PONE-D-22-05990

We would like to acknowledge all the reviewers of this manuscript for their detailed review and providing us constructive comments and professional recommendations to improve the quality of the manuscript. 

Here below, we have tried to give a point-by-point response to each inquiry by the respected editor and academic reviewers as indicated in red color.

Reviewers' comments:

Reviewer's Responses to Questions

Comments to the Author

1. Is the manuscript technically sound, and do the data support the conclusions?

Reviewer #1: Yes

Author response: Thank you

Reviewer #2: Partly

Author response: Thank you, we hope we have addressed yur issues in the revised manuscript.

2. Has the statistical analysis been performed appropriately and rigorously?

Reviewer #1: Yes

Author response: Thank you

Reviewer #2: No

Author response: Thank you for your concern, by its revised form , we hope we have made the statistical analysis process clear.

3. Have the authors made all data underlying the findings in their manuscript fully available?

Reviewer #1: Yes

Author response: Thank you

Reviewer #2: Yes

Author response: Thank you 

4. Is the manuscript presented in an intelligible fashion and written in standard English?

Reviewer #1: No

Author response: Thank you for your concern; we have revised it now and we hope you will find it comfortable. 

Reviewer #2: Yes

Author response: Thank you

5. Review Comments to the Author 

Reviewer #1: The fact that this is an institution-based study among caregivers of patient with epilepsy at public hospitals is a great strength of the investigators' approach. While this is not clearly stated, and I strongly encourage the investigators to state this clearly and underscore the contribution of this study to the overall literature. focus on important elements of the methods section, grammar, and language use. Comments are provided from here with 1-16, look there all

1. Abstract background line 26, say depression and anxiety disorders are also today’s silent health crises in primary caregivers of children and adolescents with epilepsy? Tell as some thing that you consider silent health crisis unless you must rewrite and revise the statement as well.

Author response: thank you for your point here; here we wanted to say “depression and anxiety are unforeseen among primary caregivers including primary caregivers of patients with epilepsy.”, and we have revised it in such a way in the new revsed manuscript. 

1. Abstract method part line33-34, Binary logistic regression was employed to carry out bivariate and multivariate analyses were fitted independently for both depression and anxiety. please either separate or rewrite it again.

Author response: Thank you, we have rewritten it as per your recommendation. 

2. Why nonresponse rate is high? which is around 10%

Author response: exactly it 9.5%; it is because there were participants discontinued the interview process after they agreed to give information. So, we have considered the incomplete questionnaire as a non-response since they have significant section of the questions incomplete. Another reason was unintended participants were interviewed because of technical and systematic error of selection of participants. For example, there were four participants who were not primary caregivers of patients with epilepsy; rather they were their neighbors or relatives who are not giving direct care for patients with epilepsy. These scenarios are what we have encountered during data collection. 

3. Be consistent when you write a statement especially “depression and “anxiety” or anxiety and depression”. When we see the topic, it says Depression and anxiety and their associated factors but, in abstract part of result line 38, prevalence of anxiety and depression were found to be 10.4% and 13.7% respectively. Which one is correct?

Author response: thank you, we have revised it as per your recommendation.

4. Please define it what mean by unmarried? line 39

Author response: as we have mentioned it the factors section of the result unmarried category includes those who are single, divorced, widowed and separated by their marital status. We have also added a note below the regression tables what unmarried mean in the revised manuscript. Kindly see the revised manuscript. 

5. Similar with the above one “having history of chronic medical illness” who are they chronic medical illness? do they have followed up or simply report of care giver?

Author response: thank you for your important point. We have added a statement which operationalizes what chronic medical illness means and what are the chronic medical illnesses in the operational definition section. 

6. Do you have any suggestion why duration classification with this “duration of care 6-11years”?

Author response: thank you for your interesting question; we have used this classification for duration of care from previous research conducted on this regard. 

7. Avoid such a kind of words” almost” rather put the number or percent alone.

Author response: thank you for your professional recommendation; we have revised accordingly. 

8. Be consistent either use “general anxiety disorders” or “anxiety” line 96

9. Author response: thank you for your professional recommendation; we have revised accordingly. 

10. Occupational status classification “other “is greater than housewife which is 8%, it is not advisable this number of individuals to put with term other rather put with their own classification or put remark below the table.

Author response: exactly, we have acknowledged your comment and it should not exceed the percentage of any specific categories. We have corrected it. Kindly see the revised manuscript. 

11. Line 211 what is the term” Gad” mean?

Author response: we would like to say sorry for the typing error we made; it was to mean GAD which is also inappropriate and revised accordingly in the new manuscript.

12. Line 213 “P-value “repeated two times.

Author response: thank you; we have corrected it accordingly.

13. In logistic regression the value after a point should be two digits.

Author response: thank you, we have accepted the recommendation and revised all the digits to be two after the points. 

14. On table -3 of logistic regression model, you must put the value of each variable respondents.

15. Please put confidence interval of the prevalence unless it is difficult to know and compare with others.

Author response: thank you for your significant comment here. We have added the 95% confidence intervals of the prevalence as per your recommendation.

16. It is difficult to understand the English language

Author response: we have significantly revised the English language of the paper. We hope the language of the paper is now clear for the reader. Kindly see the revised manuscript

Reviewer #2: Dear authors, I revised with interest your manuscript entitled “Depression and anxiety and their associated factors among care givers of children and adolescents with epilepsy in selected three hospitals of Amhara region, Ethiopia” for PLOS ONE, where you present a cross-sectional analysis on the risk of developing anxiety and depression in relation to caregiving activity of children with epilepsy.

In general, the manuscript is well written: the analytic plan is simple but consistent with the aims and results are clearly presented. I have few comments for your which I hope you can exploit to enhance the overall quality of your paper.

1. In general, a revision on the use of English is needed in different sections of the manuscript: in some places, I had difficulties in understanding what you meant.

Author response: we have significantly revised the English language of the paper. We hope the language of the paper is now clear for the reader. Kindly see the revised manuscript

2. Please ensure that abbreviations are given in full at the first use, abstract included.

Author response: thank you very much; we have given the full text for abbreviations in the document t its first appearance. Kindly see the revised manuscript.

3. Lines 83-90: a revision on the use of English is essential

Author response: thank you; we have revised it for English language.

4. Lines 205-209: please give means and SD for both GAD-7 and PHQ-9.

Author response: thank you for your notice; we have added mean and SD values for both PHQ-9 and GAD-7. 

5. Liens 211-258: these sections largely duplicate tables 3 and 4, and by this I mean that at least 50% of the text could be deleted. You have to reduce redundancy as it makes reading difficult: just focus on the results of the multivariable regression analysis and do not repeat OR values.

Author response: thank you; it has been corrected as per your recommendation. Kindly see the revised manuscript.

6. With regard to OR, please just refer to them as “OR” and not to AOR (I presume “adjusted” OR) or COR (unclear what does the C stand for).

Author response: With regard to COR, C stand for crude. It is to mean Crude Odds Ratio. The table incorporates both the adjusted and unadjusted (Crude) odds ratio of variable. Now, in the revised manuscript a note has been given regarding COR and AOR and we hope it is clear.

7. Table 3 and 4: I do not see the utility of the “COR” column, and my suggestion is to delete it.

Author response: there are columns showing COR values in both table 3 and 4. Kindly see the revised manuscript. 

8. An important element that is missing here is a measure of model goodness. You have the OR, however how much of anxiety/depression is explained by your predictor. Instead of using Nagelkerke pseudo R-squared, please use the C-Statistic (i.e. the area under the receiver operating curve - ROC - for the predicted versus the actual data) used to assess the whole explanatory power of the model.

Author response: As you know R2 of a model measures how well a model fits the data and is a measure of the shared variation between two or more variables. Its equivalent measure for logistic regression is the pseudo-R2. A pseudo-R2 sometimes presented alongside the area under the receiver operator characteristic (ROC) as a measure of a model’s predictive accuracy. Besides this, all the data should be binary (our data is not binary) for ROC. Hence, our study is not aimed to show prediction value rather the association of independent variables and dependent variable. However, we conducted the Hosmer-lemeshaw test and the result for both models indicated as the variables analyzed are explanatory. The p-values for models fitted for depression and anxiety were 0.73 and 0.78 respectively. We have added these values below each table in the revised manuscript.

9. Also, please include the intercept in the model and provide the -2 log likelihood difference with chi-square to test the difference between the full model and the model based on the intercept only.

Author response: thank you for your recommendation; please understand as explanation given above. Thank you

10. Please begin discussion with a paragraph stating the most relevant results. Do not repeat results (with data), but state what was found in “practical terms”. Once this is done, you can start discussing against relevant literature.

Author response: thank you for your recommendation; we have tried to put the practical finding as a starting paragraph for discussion. 

………………………………//…………………………………………..

---

## [Decision Letter · Decision Letter 1]

11 Jul 2022

Depression and anxiety and their associated factors among caregivers of children and adolescents with epilepsy in three selected hospitals in the Amhara region, Ethiopia: A cross-sectional study

PONE-D-22-05990R1

Dear Dr. BIRESAW,

We’re pleased to inform you that your manuscript has been judged scientifically suitable for publication and will be formally accepted for publication once it meets all outstanding technical requirements.

Kind regards,

Francesco Deleo, MD

Academic Editor

PLOS ONE

Additional Editor Comments (optional):

Reviewers' comments:

Reviewer's Responses to Questions

**Comments to the Author**

1. If the authors have adequately addressed your comments raised in a previous round of review and you feel that this manuscript is now acceptable for publication, you may indicate that here to bypass the “Comments to the Author” section, enter your conflict of interest statement in the “Confidential to Editor” section, and submit your "Accept" recommendation.

Reviewer #1: All comments have been addressed

Reviewer #2: All comments have been addressed

2. Is the manuscript technically sound, and do the data support the conclusions?

Reviewer #1: Yes

Reviewer #2: Yes

3. Has the statistical analysis been performed appropriately and rigorously? 

Reviewer #1: Yes

Reviewer #2: Yes

4. Have the authors made all data underlying the findings in their manuscript fully available?

Reviewer #1: Yes

Reviewer #2: Yes

5. Is the manuscript presented in an intelligible fashion and written in standard English?

Reviewer #1: Yes

Reviewer #2: Yes

6. Review Comments to the Author

Reviewer #1: That is excellent all the comments that i raised is addressed and got adequate information from your research especially the revised one.

Reviewer #2: Dear authors, I revised the new version of your manuscript on depression and anxiety in caregivers of children with epilepsy in Ethiopia. I think the revisions are satisfactory and have no other ones for you.

7. PLOS authors have the option to publish the peer review history of their article (what does this mean?). If published, this will include your full peer review and any attached files.

Reviewer #1: **Yes: **Wondale Getinet Alemu(PhD Fellow ,Flinders University,South Australia)

Reviewer #2: No

---

## [Editor Report · Acceptance letter]

15 Jul 2022

PONE-D-22-05990R1 

Depression and anxiety and their associated factors among caregivers of children and adolescents with epilepsy in three selected hospitals in Amhara region, Ethiopia: A cross-sectional study 

Dear Dr. BIRESAW:

I'm pleased to inform you that your manuscript has been deemed suitable for publication in PLOS ONE. Congratulations! Your manuscript is now with our production department. 

Kind regards, 

on behalf of

Dr. Francesco Deleo 

Academic Editor

PLOS ONE